# Epidemiology and clinical characteristics of viral infections in hospitalized children and adolescents with cancer in Lebanon

Sarah Chamseddine[1☯], Ahmad Chmaisse[1☯], Imad Akel[1], Zeinab El Zein[1,2], Suzan Khalil[1], Sarah Abi Raad[1], Antoine Khati[1], Hiba Ghandour[1], Sarah Khafaja[1,2], Magda Haj[1], Miguel Abboud[2,3], Rami Mahfouz[1,4], George Araj[1,4], Hassan Zaraket[1,5], Rima Hanna-Wakim[1,2,6], Samar Muwakkit[2,3], Ghassan Dbaibo[1,2,6]*

1 Center for Infectious Diseases Research, American University of Beirut, Beirut, Lebanon, 2 Department of Pediatrics and Adolescent Medicine, American University of Beirut Medical Center, Beirut, Lebanon, 3 Department of Pediatrics and Adolescent Medicine, Children's Cancer Center of Lebanon, American University of Beirut Medical Center, Beirut, Lebanon, 4 Department of Pathology and Laboratory Medicine, American University of Beirut Medical Center, Beirut, Lebanon, 5 Department of Experimental Pathology, Immunology and Microbiology, Faculty of Medicine, Beirut, Lebanon, 6 Division of Pediatric Infectious Diseases, Department of Pediatrics and Adolescent Medicine, American University of Beirut Medical Center, Beirut, Lebanon

☯ These authors contributed equally to this work.
* gdbaibo@aub.edu.lb

## Abstract

### Background

Viral infections in children and adolescents with malignancy are commonly encountered and have a significant impact on morbidity and mortality. Studies and epidemiological data regarding viral infections in children with cancer in developing countries are lacking. This retrospective cohort study aims to assess the burden of viral infections in children and adolescents with cancer, by assessing prevalence, risk factors, as well as morbidity and mortality of common viruses over a period of 8 years.

### Methods and findings

Medical records of cancer patients treated at the Children Cancer Center of Lebanon were reviewed and 155 participants under the age of 21 were identified with at least one documented viral infection during the period from July 2009 to November 2017. This subset included 136 participants with active malignancy and 19 participants with a history of cancer who underwent hematopoietic stem cell transplantation [HSCT] and were in remission; the latter group was analyzed separately. Information regarding participant characteristics, hospital course, and complications were obtained. Associations between viral infections and certain factors were assessed. In the cohort, 64% were male, 81% were Lebanese. In participants with active malignancy, 90% received chemotherapy in the 6 months preceding the viral infection episode, 11% received radiotherapy. 51% of participants were neutropenic at the time of viral detection, and 77% were lymphopenic. 17% experienced a bacterial co-infection, and 3 experienced a viral co-infection. Among 162 viral infection episodes,

their request to the Institutional Review Board (IRB) at the American University of Beirut (Human Research Protection Program (HRPP)). Contact via irb@aub.edu.lb.

**Funding:** The author(s) received no specific funding for this work.

**Competing interests:** The authors have declared that no competing interests exist.

clinically diagnosed skin infections, mainly herpes simplex virus type 1 and varicella-zoster virus, were the most common [44% of cases]. These were followed by laboratory-proven systemic herpes infections: cytomegalovirus [14%] and Epstein-Barr virus [6%]. Respiratory viruses: influenza and respiratory syncytial virus, accounted for 9% and 4%, respectively, whereas rotavirus represented 11% and BK virus represented 3% of cases. Acute lymphocytic leukemia was the most prevalent neoplasia [57%]. Fever was the most common presenting symptom [55%] and febrile neutropenia was the reason for admission in 24% of cases. The mean length of stay was significantly longer in participants with cytomegalovirus infections and significantly lower in rotavirus infection. Admission to the ICU occurred in 9%, complications in 8%, and mortality in 5%. Participants with viral infections post-HSCT were noted to have a significantly longer length of hospital stay compared to non-HSCT participants, with no other significant differences in clinical course and outcome. The study was limited by its retrospective nature and by the late introduction and underuse of multiplex PCR panels, which may have led to underdiagnosis of viral infections.

## Conclusions

Viral infections were prevalent in our sample of cancer patients and may have contributed to morbidity and mortality. Newly available viral diagnostics are likely to vastly increase the number and scope of detectable viral infections in this population. Prospective studies using multiplex PCR technology with systematic testing of patients will be more helpful in defining the burden of viral infections. Furthermore, efforts at antimicrobial stewardship would benefit from the identification of viral causes of infection and limit the unnecessary use of antibiotics in the pediatric cancer population.

## Introduction

Due to the nature of their disease and related therapy, cancer patients are at high risk for developing various infections [1–5]. These infectious processes have a significant impact on morbidity and mortality, with higher standard mortality ratios compared to other causes of mortality [2, 3]. Hence, prophylaxis covering a broad range of bacterial, fungal, and viral agents is commonly administered according to the type of cancer and treatment [6]. As for viral infections, treatments range from supportive therapy to newer anti-viral drugs that are expensive and of limited availability, especially in developing countries. Thus, viral infections are less commonly investigated, and prophylaxis against a limited range of these infections is considered [5]. Similarly, febrile neutropenia protocols and guidelines support the early use of antibiotics [7], although literature has reported viral infections to constitute 35% to 55% of all admissions with febrile neutropenia in the pediatric population [8–10]. Therefore, understanding the true burden of viral infections and having the ability to diagnose them early during the febrile illness would have a positive impact on antimicrobial stewardship efforts without compromising patient outcome.

Viral infections can affect several systems with variable clinical courses [11]. Despite the advent of newer, cheaper, and more widely available biochemical and molecular techniques, these infections are still challenging to diagnose [8]. The prevalence of viral infections in pediatric malignancy has been documented in certain settings, reaching 50% of all infectious episodes in a cohort of 610 Acute Lymphocytic Leukemia (ALL) patients in Greece [12]. A

Finnish study in ALL patients also confirmed a higher incidence of viral infections in cancer patients, as compared to controls [13]. Another study from Austria demonstrated that the prevalence of viral respiratory infections in children with cancer (53%) was double that of immunocompetent hosts (25%), and those receiving hematopoietic stem cell transplants were even more predisposed to developing viral infections (81%) [14]. Other studies assessing the epidemiology of viral infections have found variable prevalence, but high morbidity has been shown consistently in patients with cancer [15].

Studies and epidemiological data regarding viral infections in children and adolescents with cancer from developing countries are lacking. In a previous prospective study on respiratory infections, we showed a high burden of respiratory viral infections and a high incidence of coinfections in the pediatric cancer patients [5]. There is a need to evaluate the burden that viral infections impose on this susceptible population in countries with limited resources. Knowledge of the epidemiology of viral infections in this population could provide clues on how to better prioritize, diagnose, treat, and prevent them in the future. Moreover, given the increasing resistance of bacterial pathogens and the limited arsenal of new antibiotics, this knowledge could also help to guide physicians in the global efforts of antimicrobial stewardship. In the current study, we have reviewed our experience at the Children's Cancer Center of Lebanon (CCCL) over a period of 8 years in order to assess the burden of viral infections in children and adolescents with cancer and/or hematopoietic stem cell transplantation [HSCT] in a resource-limited setting and identify priorities for capacity building in countries with similar settings.

## Materials and methods

### Study design

The American University of Beirut Medical Center (AUBMC) is a tertiary medical care center located in Beirut, Lebanon that accomodates around 9500 pediatric admissions annually. It includes the independently goverened Children's Cancer Center of Lebanon (CCCL), which is run by a non-profit association affiliated to St. Jude's Children's Research Hospital (SJCRH) in Memphis, Tennessee. CCCL consists of 18 inpatient beds and 10 outpatient cubicles, and admits approximately 1080 pediatric cancer patients annually, with 120 new patients admitted per year. Patients between the ages of 0 and 21 years of age are referred to CCCL from all regions in Lebanon in addition to neigboring countries.

We performed a retrospective cohort study in which medical records of patients treated at CCCL were reviewed. The Institutional Review Board (IRB) at the American University of Beirut approved this proposal (PED.GD.07) and waived the requirement for informed consent due to the retrospective nature of our study. Data was only anonymized at the analysis phase and thereafter, as the Medical Record Numbers (MRNs) were used to access patients' charts and retrieve the data during the collection phase. The study included participants under 21 years of age with active cancer admitted to AUBMC with a confirmed diagnosis of at least one viral infection, during the period from July 2009 to November 2017. Participants were identified using two Electronic Health Record (EHR) search approaches. Table 1 was used to guide the search in the first approach. The first three columns were prepared beforehand to include commonly encountered viral infections in children and adolescents with cancer, along with the commercial used for each virus at our institution. EHR was then used to search for these viruses in the laboratory records of children and adolscenets with cancer; participants with at least one virus mentioned in their laboratory records were included. These viruses were detected/considered to be the source of infection when commercial PCR kits, antibody detection, and/or antigen detection gave a positive result from samples taken from all body sources

**Table 1. Commonly encountered viruses in children and adolescents with cancer, with corresponding detection methods.**

| Virus | Methods of Detection (MOD) | Commercial Kit | Identified Cases through MODs | Sample Source |
|---|---|---|---|---|
| Adenovirus | PCR | Qiagen | 3 | Nasal: 1 |
| | | | | Blood: 1 |
| | | | | Stool: 1 |
| BK Virus | PCR | FTD | 5 | Urine: 5 |
| Coronavirus | PCR | Respiratory panel-FilmArray | 1 | Nasopharyngeal: 1 |
| Cytomegalovirus | PCR | Qiagen & Roche | 22 | Blood: 22 |
| | IgM | Architect Abbott | 1 | Blood: 1 |
| Epstein–Barr virus | PCR | Qiagen | 10 | Blood: 10 |
| | IgM | Architect Abbott | 0 | - |
| Hepatitis A | IgM | Architect Abbott | 2 | Blood: 2 |
| Hepatitis B | PCR | Roche | 1 | Blood: 1 |
| Hepatitis C | PCR | Roche | 2 | Blood: 2 |
| Herpes Simplex Virus 1,2 | PCR | Qiagen | 10 | Skin Lesions: 10 |
| Human herpesvirus 6 | PCR | Qiagen | 0 | - |
| Human Immunodeficiency Virus | PCR | Cepheid & Roche | 0 | - |
| | Antibodies to HIV-1 and -2 | Architect Abbott | 0 | - |
| Human Papillomavirus | PCR | Cepheid | 0 | - |
| Influenza | PCR | 1) Cepheid 2) Respiratory Panel-FilmArray | 0 | - |
| | Antigen | BD Veritor System | 15 | Nasopharyngeal: 15 |
| Measles | PCR | Referred-out | 0 | - |
| | IgM | Euroimmun | 0 | - |
| Mumps | PCR | Referred-out | 0 | - |
| | IgM | Euroimmun | 1 | Blood: 1 |
| Norovirus | PCR | Cepheid | 0 | - |
| Parvovirus | PCR | Fast-track | 2 | Blood: 2 |
| Respiratory Syncytial Virus | PCR | 1) Cepheid 2) Respiratory Panel-FilmArray | 0 | - |
| | Antigen | SAS RSV Alert | 7 | Nasal: 7 |
| Rotavirus | Antigen | SD BIOLINE Rotavirus | 19 | Stool: 19 |
| Varicella | PCR | Qiagen | 0 | - |
| | IgM | Euroimmun | 0 | - |

[blood, cerebrospinal fluid, sputum, deep tracheal aspirate, urine, stools, swabs...]. The other approach was based on discharge diagnoses codes of common clinically diagnosed viral infections. These codes were considered to be a reliable method to screen patients who might have presented and were later discharged with certain viral diagnosis. The codes included: Code 052 for varicella infections, Code 053 for herpes zoster infections, Code 054 for herpetic infections, and Code 055 for measles infections. Encounters of children and adolescents with cancer on EHR were searched for these codes; participants with at least one code present in their encounters' discharge diagnoses were included.

## Data collection

Data collected included the following information: basic demographic and epidemiological charateristics (age, gender, nationality), primary diagnosis of malignancy, time since diagnosis, chemotherapy, history of radiotherapy, history of intravenous immunoglobulin infusion, and presence of neutropenia (<1500/microL) and/or lymphopenia (<1500/microL). Time since

chemotherapy was divided into ongoing/within six months or more than six months after the last cycle. For participants who had received a HSCT, the date of HSCT along with the presence of allogenic engraftment was documented.

### Hospital admission and course of infection

Hospital data regarding the admission were collected and analyzed. This data included: admission diagnosis, length of hospitalization, date of viral infection(s) identification, virus(es) involved, methods of laboratory confirmation, bacterial and/or viral co-detection, and recurrence of infection. If the participant had at least one virus detected during the admission, it was considered a single viral infection episode regardless of the number of viruses detected simultaneously. Participants who had complete resolution of the infection, whether clinically and/or laboratory-proven, with recurrence of the same viral infection within one year were included as separate cases. Initial presentation of viral infection was also examined and symptoms on presentation were recorded. The clinical outcomes identified included length of hospital stay, intensive care unit (ICU) admissions, intubations, complications, and death. Complications included pleural or pericardial effusions, ascites, respiratory, renal, heart or liver failure, and disseminated intravascular coagulation (DIC).

### Special considerations

For participants who were recurrently admitted with the same viral infection, only the first admission with the proven infection was reported as a separate case. In subsequent admissions, where viral load (in cytomegalovirus, Epstein-Barr virus, and adenovirus infections) or clinical manifestations were still present, we only searched for complications and treatment modalities pertaining to the primary infection.

### Statistical analysis

Data were collected on a Case Report Form and later analyzed with the Statistical Package for Social Sciences (SPSS) program, version 23.0 for Windows [(IBM, Armonk, NY). Univariate and multivariate analysis (Fisher's tests, Chi-Squared test) were used to determine associations between certain factors (cancer type, recent chemotherapy, history of radiotherapy, neutropenia/lymphopenia at time of diagnosis) and viral infections, and between viral infections and outcomes.

## Results

### Participant characteristics and detected viruses

The total number of pediatric cancer cases enrolled at CCCL within the timeframe of our study was 778 cases, distributed as following: 257 leukemia cases (33%), 119 lymphoma cases (15%), and 402 solid tumor cases (52%). The number of viral infection episodes, with at least one detected virus, was 162; viral co-detection occurred in three of them. Participant characteristics at the time of the viral infection episode are presented in Table 2. These episodes occurred in 136 admitted participants (17% of total participants enrolled), where some participants had more than one episode of viral infection. 27 episodes of viral infection were also reported in 19 (2%) admitted participants who were in remission post-HSCT. Table 1 lists the number of detected viruses through MOD, along with the source of sample for laboratory detected viruses. Table 3 summarizes risk factors and laboratory findings and Table 4 the presenting symptoms observed with the encountered viruses.

**Table 2. Socio-demographic characteristics, laboratory findings, and malignancy types of participants at times of viral infection episodes.**

| Characteristics | Participants at time of viral infection episodes [%], N = 162 |
|---|---|
| **Gender** | |
| *Male* | 104 [64.2] |
| *Female* | 58 [35.8] |
| **Age** | |
| *Less than 1 year* | 9 [5] |
| *1 to 2 years* | 9 [5] |
| *2 to 5 years* | 48 [30] |
| *5 to 10 years* | 39 [24] |
| *10 to 15 years* | 36 [16] |
| *15 to 21 years* | 21 [13] |
| **Chemotherapy in the last 6 months** | 145 [89.5] |
| **Radiotherapy** | 17 [10.5] |
| **Neutropenia** | 83 [51.2] |
| **Lymphopenia** | 124 [76.5] |
| **Bacterial co-detection** | 27 [16.7] |
| **Viral co-detection** | 3 [1.8] |
| **Malignancy types** | |
| *Leukemia* | 92 [56.8] |
| *Solid tumors* | 49 [30.2] |
| *Lymphoma* | 21 [13.0] |

Febrile neutropenia was the reason for admission in a total of 38 cases. Of these, 12 (31%) were found to have herpes simplex virus (HSV) infection. Nine participants (23%) were cytomegalovirus (CMV) positive by PCR of the blood, and five participants (13%) had varicella-zoster virus (VZV) infections. Only 6 of the 38 cases (1 EBV case, 2 HSV, 1 VZV, 1 influenza, 1 CMV) had bacterial co-detection, either in blood or urine. All participants were undergoing chemotherapy, and 74% had underlying leukemia.

## Documented viral infections per viral category

The identified viral infections were classified into four main categories: herpesvirus infections, respiratory viral infections, gastrointestinal viral infections, and BK viral infections. Clinical characteristics of admitted participants with viral infections, along with the presenting

**Table 3. Risk factors and laboratory findings of different viral infections in children and adolescents with cancer.**

| | CMV | EBV | VZV | HSV | Influenza virus | RSV | Rotavirus | BK virus | p-value[*] |
|---|---|---|---|---|---|---|---|---|---|
| | N = 23 | N = 10 | N = 43 | N = 29 | N = 15 | N = 7 | N = 19 | N = 5 | |
| | (n [%]) | (n [%]) | (n [%]) | (n [%]) | (n [%]) | (n [%]) | (n [%]) | (n [%]) | |
| **Chemotherapy in the last 6 months** | 23 [100] | 8 [80.0] | 35 [81.4] | 29 [100] | 15 [100] | 6 [85.7] | 16 [84.2] | 5 [100] | **0.013** |
| **Radiotherapy** | 0 [0.0] | 1 [10.0] | 8 [18.6] | 2 [6.9] | 0 [0.0] | 1 [14.3] | 4 [21.1] | 1 [20.0] | 0.105 |
| **Neutropenia** | 15 [65.2] | 6 [60.0] | 19 [44.2] | 17 [58.6] | 8 [53.3] | 4 [57.1] | 6 [31.6] | 2 [40.0] | 0.409 |
| **Lymphopenia** | 19 [82.6] | 7 [70.0] | 33 [76.7] | 23 [79.3] | 14 [93.3] | 3 [42.9] | 15 [78.9] | 5 [100] | 0.276 |

[*]$H_0$ tested: There is no association between risk factors (chemotherapy, radiotherapy)/laboratory finding (neutropenia and lymphopenia) and particular viruses, significant at p<0.05. Fisher's exact test was used when expected count was less than 5.

**Table 4. Presenting Symptoms associated with different types of viral infections in children and adolescents with cancer.**

|  | CMV | EBV | VZV | HSV | Influenza virus | RSV | Rotavirus | BK virus |
|---|---|---|---|---|---|---|---|---|
|  | N = 23 | N = 10 | N = 43 | N = 29 | N = 15 | N = 7 | N = 19 | N = 5 |
|  | (n [%]) | (n [%]) | (n [%]) | (n [%]) | (n [%]) | (n [%]) | (n [%]) | (n [%]) |
| **Fever** | 21 [91.3] | 8 [80.0] | 20 [4.5] | 13 [44.8] | 11 [73.3] | 5 [71.4] | 10 [52.6] | 2 [40.0] |
| **Cough** | 6 [26.1] | 2 [20.0] | 6 [14.0] | 6 [20.7] | 11 [73.3] | 5 [71.4] | 1 [5.3] | 0 [0.0] |
| **Rhinorrhea** | 5 [21.7] | 1 [10.0] | 4 [9.3] | 1 [3.4] | 9 [60.0] | 4 [57.1] | 0 [0.0] | 0 [0.0] |
| **Gastrointestinal symptoms[+]** | 9 [39.1] | 6 [60.0] | 5 [11.6] | 6 [20.7] | 3 [20.0] | 2 [28.6] | 19 [100] | 1 [20.0] |

[+] Gastrointestinal symptoms included diarrhea, vomiting, constipation or abdominal pain.

symptoms and the corresponding clinical outcomes are reported in Tables 5–7, respectively. All participants who had been in remission post-HSCT are discussed separately.

**Herpesviruses infections.** Herpesviruses infections accounted for 64% of all identified viral infections and included EBV, CMV, HSV, and VZV.

All reported EBV and CMV infections were laboratory-confirmed via blood PCR, with the exception of 1 participant with positive CMV IgM. EBV and CMV most commonly presented with fever and gastrointestinal symptoms, with fever present in 91% of all CMV infections. This proportion was highest among all the viruses studied (p = 0.004). Furthermore, of the 23 participants with CMV, three were diagnosed with pneumonitis based on the clinical picture and/or radiologic evidence, and one participant developed CMV retinitis. The latter participant had the longest duration of infection at 580 days with no documentation of viral clearance by PCR in the medical record. Subsequently participants with CMV infection had the longest mean length of hospital stay. The mean duration of viremia from first detection to first negative was 122 days and 71.9 days in EBV and CMV respectively. Viral reactivation within the first year after negative PCR was common within this group as well, occurring in 20% of participants with EBV and 35% of participants with CMV. The mortality rate of EBV was highest at 20% compared to all other identified viruses, however the 2 participants had fungal or bacterial co-infections and ultimately developed DIC and hemophagocytic lymphohistiocytosis (HLH) respectively.

**Table 5. Clinical outcomes of viral infections in children and adolescents with cancer.**

| Mean [±SD] | | | | | | | | | |
|---|---|---|---|---|---|---|---|---|---|
|  | CMV | EBV | VZV | HSV | Influenza virus | RSV | Rotavirus | BK virus | p-value* |
|  | N = 23 | N = 10 | N = 43 | N = 29 | N = 15 | N = 7 | N = 19 | N = 5 |  |
|  | (n [%]) | (n [%]) | (n [%]) | (n [%]) | (n [%]) | (n [%]) | (n [%]) | (n [%]) |  |
| **Mean length of stay** | 17.39 [±11.0] | 15.1 [±11.9] | 8.5 [±7.7] | 9.7 [±11.0] | 8.1 [±4.5] | 9.0 [±5.9] | 4.3 [±3.7] | 34 [±62.1] | **0.001** |
| n [%] | | | | | | | | | |
|  | CMV | EBV | VZV | HSV | Influenza virus | RSV | Rotavirus | BK virus | p-value* |
|  | N = 23 | N = 10 | N = 43 | N = 29 | N = 15 | N = 7 | N = 19 | N = 5 |  |
|  | (n [%]) | (n [%]) | (n [%]) | (n [%]) | (n [%]) | (n [%]) | (n [%]) | (n [%]) |  |
| **ICU admission** | 5 [21.7] | 2 [20.0] | 2 [4.7] | 0 [0.0] | 1 [6.7] | 2 [28.6] | 2 [10.5] | 0 [0.0] | **0.029** |
| **Intubation** | 3 [13.0] | 1 [10.0] | 2 [4.7] | 0 [0.0] | 1 [6.7] | 0 [0.0] | 0 [0.0] | 0 [0.0] | 0.329 |
| **Complications[+]** | 3 [13.0] | 1 [10.0] | 3 [7.0] | 1 [3.4] | 2 [13.3] | 1 [14.3] | 1 [5.3] | 0 [0.0] | 0.748 |
| **Death** | 2 [8.7] | 2 [20.0] | 2 [4.7] | 0 [0.0] | 1 [6.7] | 1 [14.3] | 0 [0.0] | 0 [0.0] | 0.152 |

* $H_0$ tested: There is no association between clinical outcomes (length of stay, ICU admission, intubation, complications, death) and particular viruses, significant at p<0.05. Fisher's exact test was used when expected count was less than 5.

[+] Complications included coma, DIC, ascites, pleural or pericardial effusions, liver failure, cardiac failure, renal failure, and/or respiratory failure.

**Table 6. Distribution of viral infections in different malignancy groups.**

|  | CMV | EBV | VZV | HSV | Influenza virus | RSV | Rotavirus | BK virus |
|---|---|---|---|---|---|---|---|---|
|  | N = 23 | N = 10 | N = 43 | N = 29 | N = 15 | N = 7 | N = 19 | N = 5 |
|  | (n [%]) | (n [%]) | (n [%]) | (n [%]) | (n [%]) | (n [%]) | (n [%]) | (n [%]) |
| Leukemia | 15 [60.9] | 6 [60.0] | 21 [50.0] | 23 [79.3] | 9 [60.0] | 2 [28.6] | 8 [42.1] | 0 [0.0] |
| Lymphoma | 4 [17.4] | 4 [40.0] | 8 [18.2] | 2 [6.9] | 2 [13.3] | 0 [0.0] | 0 [0.0] | 0 [0.0] |
| Solid tumors | 5 [21.7] | 0 [0.0] | 14 [31.8] | 4 [13.8] | 4 [26.7] | 5 [71.4] | 11 [57.9] | 5 [100] |

For participants with VZV and HSV, the majority of cases (86%) were clinically diagnosed and not documented by laboratory testing, with 10 cases of HSV infection confirmed by PCR of skin lesions. VZV was the most frequently encountered virus in the overall cohort.

Of the 29 participants diagnosed with HSV infection, 17 (59%) were admitted for another reason and the lesions were incidentally noted. The participants had all received chemotherapy within six months of the viral infection. As for participants with VZV infections, 26 (60%) presented with a primary chickenpox infection, while 17 (40%) presented with a reactivated VZV infection. Rash with fever was the most common presenting finding (32% of cases). None of the participants with HSV had complicated or disseminated disease, though 24 participants (82%) received intravenous anti-viral therapy with Acyclovir. In contrast, 2 participants with VZV were suspected of having disseminated VZV infection and eventually developed respiratory failure and died. Both had received chemotherapy within one week of infection and did not have any underlying comorbidities.

Within this cohort of participants found to have herpesvirus infections, 3 viral co-infections (CMV/HepB, EBV/HSV, VZV/Rotavirus) were documented, with none of the participants experiencing additional complications. Four participants within the same cohort had underwent a HSCT in the past, but three were in relapse (1 CMV, 2 VZV), and one was still undergoing radiotherapy for his underlying malignancy (1 HSV) at the time of viral infection.

**Respiratory infections.** Respiratory viral infections accounted for 13% of the identified viral infections in this cohort. The most frequent respiratory infections identified included influenza virus (15 participants, 68%) and RSV (7 participants, 32%), all confirmed through direct antigen testing of nasal, nasopharyngeal, and pharyngeal swab samples. To note, the proportion of participants presenting with cough and/or rhinorrhea was highest in this group (p = 0.00). One case of coronavirus was also identified by a multiplex PCR respiratory ultrapanel, the only virus to be detected by such panel in this cohort. Only one viral co-detection was reported with influenza and adenovirus. One participant with RSV had previously received allogenic HSCT 2 years prior to the infection but had relapsed within that time interval.

The mortality rate amongst this cohort was relatively high at 9% and included an influenza participant with underlying common variable immune deficiency (CVID) and lymphoma who developed a superimposed pneumonia that was later complicated by acute respiratory distress syndrome (ARDS). He had no documentation of vaccination status in the medical record.

**Table 7. Comparison of outcomes from viral infection episodes between HSCT and non-HSCT recipients.**

|  | HSCT [N = 27] | Non- HSCT [N = 162] | p-value |
|---|---|---|---|
| Mean length of hospital stay [n days] | 21.4 days | 10.3 days | 0.001 |
| ICU admission [n (%)] | 6 (22%) | 15 (9%) | 0.167 |
| Complications [n (%)] | 4 (15%) | 15 (9%) | 0.341 |
| Death [n (%)] | 3 (11%) | 19 (6%) | 0.296 |

The second mortality was a participant with Down syndrome and leukemia who developed candidemia and acquired RSV that was treated with Palivizumab and ribavirin.

Seasonality of RSV and influenza infections portrayed a peak rate (55%) occurring in December-February (Fig 1), which coincides with the typical peak activity of these viruses in Lebanon.

**Gastrointestinal infections.** Gastrointestinal (GI) infections accounted for 12% of identified viral infections in this population. These included 19 rotavirus cases.

The age range of participants with rotavirus was 5 months to 17 years. All 19 cases, identified by direct antigen testing from stool specimens, had GI symptoms at presentation, including diarrhea and/or vomiting, a proportion higher than any other group (p = 0.00). The mean length of hospital stay was 4.2 days, which was significantly shorter than other viruses (p = 0.033). One participant with rotavirus infection developed complicated disease with pancreatitis. Vaccination records were not available in the medical records. Rotavirus was co-detected with VZV in one participant who had received an autologous HSCT four months prior to infection followed by relapse.

**BK Virus infections.** Five participants developed hemorrhagic cystitis and were found to have BK virus infection by PCR of the urine. All 5 participants had received chemotherapy within the last 6 months and had lymphopenia at the time of presentation. All 5 had been diagnosed with malignancy within the year preceding the infection. All presented with urinary symptoms along with fever in three of the five participants. None had a complicated course of illness, and no deaths were encountered. Participants with BK virus had the longest mean duration of hospital stay compared to other viruses, but this difference was not statistically significant (p = 0.43). The duration of infection until negative testing was an average of 31.6 days (± 26). No antiviral treatment was administered for these participants.

**Other viral infections.** Twelve other participants were identified with viruses that did not fit under any of the categories above and could not be analyzed as one group due to innate differences in characteristics. This group included infections with adenovirus (3 cases), parvovirus (2 cases), hepatitis A virus (2 cases), hepatitis C virus (2 cases), mumps virus (2 cases), and hepatitis B virus (1 case, previously mentioned). Hepatitis C, parvovirus, and adenovirus were detected via PCR, while all other viruses were detected by serologies with the exception of 1 clinically diagnosed Mumps infection where the participant was reportedly vaccinated.

Amongst this group of participants, only one with adenovirus had complicated disease leading to multi-organ failure and death. He had no underlying comorbidities or bacterial co-infections but expired before any treatment was initiated. He was found to be both neutropenic and lymphopenic at the time of presentation and had received chemotherapy in the preceding week.

For the participants with hepatitis C virus, neither had received any chemotherapy recently nor had any history of documented hepatitis C infection. No information on sexual activity

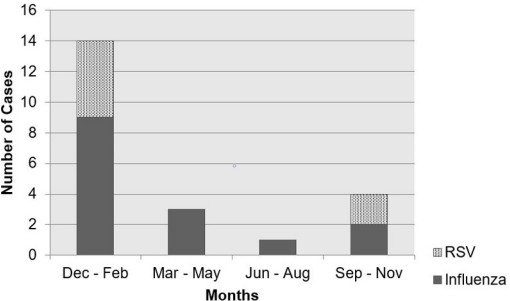

**Fig 1. Seasonality of respiratory viral infections.**

was documented. One participant had a history of several blood transfusions prior to and during the admission and had underlying liver fibrosis. As reported earlier, 1 CMV/Hepatitis B co-detection, and 1 adenovirus/influenza co-detection were reported. As for parvovirus, the two participants with parvovirus infection had characteristic disease with fever and rash. Both had received chemotherapy in the preceding week and were lymphopenic.

## Viral infections in relation to the underlying malignancy

In participants with a documented viral infection, the majority had leukemia as the underlying malignancy (57%), followed by solid tumors (30%) and lymphoma (13%). Distribution of viral infections in the different malignancy groups are presented in Table 6. With the exception of RSV, rotavirus and BK virus which were most commonly identified in participants with solid tumors, the remaining viruses were most commonly identified in participants with leukemia. To note, the differences in distribution of viruses among the different cancer groups were found to be statistically significant [P<0.001].

## Documented viral infections in HSCT participants

Out of 23 study participants who underwent HSCT as part of their treatment course (15 allogeneic, 8 autologous), 19 participants did not relapse post-transplant (12 allogeneic, 7 autologous). In this latter group, 27 viral infection episodes were documented, with one occurring as a co-infection (CMV and EBV). Viral infections were detected within the first six weeks following HSCT in 21 episodes (78%). In 4 of the 21 participants, the viral infection was detected before engraftment took place.

Distribution of infections was as follows: 13 CMV (46%), 1 EBV (4%), 5 VZV (17%), 2 HSV (7%), 1 rotavirus (4%), 1 adenovirus (4%), 4 BK virus (14%), and 1 parvovirus infection (4%). The proportions of CMV (p = 0.0001) and BK virus (p = 0.001) were significantly larger compared to the non-HSCT cancer group. Fig 2 highlights the distribution of viruses in participants with or without HSCT. To note that 70% of VZV and HSV infections occurred within 6 weeks of the transplant.

Neutropenia was present in 25% of the cases, and lymphopenia in 79%. The average length of hospital stay was 21 days (±15.5 days), with the longest stay being 61 days in the participant with CMV and EBV co-detection. A comparison between the outcomes of HSCT and non-HSCT participants is presented in Table 7. The length of hospital stay was significantly longer in HSCT participants (p = 0.001). Of these participants, 6 were admitted to the ICU (4 CMV, 1 EBV, 1 parvovirus) and 4 developed complications (3 CMV, 1 parvovirus), of which 3 (75%)

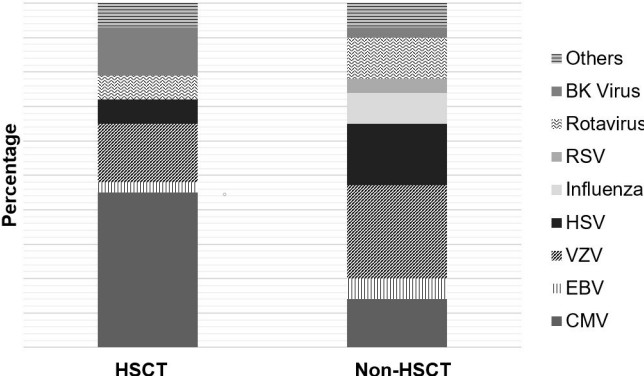

**Fig 2. Distribution of viruses in participants with or without HSCT.**

had a bacterial co-infection (p = 0.034). Three participants died during their hospital stay at a mortality rate of 11%. The first participant developed parvovirus viremia during his admission for HSCT, and subsequently developed systemic inflammatory response syndrome (SIRS) and died after cardiopulmonary arrest. He had no concomitant viral or bacterial co-detection. The two other participants developed a CMV infection shortly following their HSCT. The first developed the infection after engraftment took place and had tested positive for *E. coli* MDR in urine and *Toxoplasma gondii* DNA in the CSF. The second had concomitant Acinetobacter bacteremia, and contracted the infection before engraftment took place.

To note, five HSCT participants developed graft-versus-host disease (GVHD) following their transplant, but none of these had a complicated course with their viral infections. The cases were distributed relatively equally among all five viral categories (two VZV cases, one BK virus case, and one case with co-detection of CMV and rotavirus).

## Discussion

This study aimed to provide an overview of the burden of viral infections in children and adolescents with cancer in a resource-limited setting were not all patients are tested routinely for viruses unless there's a high clinical suspicion. Herpes simplex virus type 1 and varicella-zoster virus were the most commonly encountered viruses, and acute lymphocytic leukemia was the most prevalent neoplasia in infected participants. Of the participants with a viral infection episode, 90% had received chemotherapy in the preceding 6 months. Half of the participants were neutropenic at the time of viral detection, and febrile neutropenia was the reason for admission in a quarter of participants. The mean length of stay was significantly longer in participants with CMV infections and significantly lower in rotavirus infection. Admission to the ICU occurred in 9%, complications in 8%, and mortality in 5%.

### Herpesviruses infections

Not surprisingly, herpesviruses were the most common of all infections (64%), 42% of which were caused by VZV. Herpesviruses are very common causes of viral infections in immunocompetent as well as in immunocompromised patients, and pose high risk of morbidity and mortality in immunocompromised patients [16, 17]. Their ability to produce latency and to reactivate when immunity weakens as a result of the underlying malignancy or the associated chemotherapy largely explains their predominance. For all documented herpesvirus infections, leukemia was the most common underlying malignancy, and 90% had received chemotherapy less than 6 months before the infection.

Herpesviruses were also the most common detected viruses in the participants presenting initially for febrile neutropenia (54%). A recently conducted study reviewing causes of febrile neutropenia reported 10% of cases positive for a herpesvirus, similar to the rate of positive bacterial detection (13%) in the same cohort [18].

CMV is notorious for being associated with infections in immunocompromised hosts [19]. CMV can present as asymptomatic viremia, or cause a variety of end-organ diseases including pneumonia, retinitis, enteritis, hepatitis, and/or central nervous system infections [20, 21]. The most common presentation of participants with CMV infection in our study, as reported in the literature [17], was fever, where 91% of the infected participants had fever at initial presentation. EBV on the other hand, is highly prevalent in different populations but usually remains asymptomatic and subclinical with rare episodes of reactivation [22]. EBV can affect virtually any organ system and has been associated with diverse disease manifestations such as pneumonia, myocarditis, and pancreatitis among others [23], although the majority of the EBV cohort had uncomplicated disease. HSV-1 is not uncommon among immunocompetent hosts as well,

with 67% of the population being infected [24]. For half the participants, herpetic lesions were noticed incidentally during the hospital stay rather than being the main reason for admission. It is unclear if the infections were primary infections or reactivation of previously acquired dormant HSV-1 infections, but the former is less likely because primary HSV infections in immunocompromised children are likely to be severe.

The complication rate among herpesviruses was highest in CMV, and the mean length of hospital stay was significantly longer compared to other viruses. Risk factors for complicated CMV disease have been identified in the literature, including leukocytosis, mechanical ventilation, and delayed treatment [25, 26]. EBV was associated with the highest mortality rate, though this could be attributable to other factors, mainly a high bacterial co-infection rate in these participants. VZV mortality on the other hand, was due to disseminated infection and multi-organ failure. Comparable to our VZV mortality of 5%, a study published by Brown et al in 2016 reported 13.6% of pediatric cancer patients with VZV infections required critical care and 3.4% died due to disseminated infection [27]. Although studies have linked lymphopenia to increased mortality in VZV [28], none of the lymphopenic participants in our cohort with VZV infection developed any complications.

## Respiratory infections

The second most common category of viruses was those associated with respiratory infections. Several studies have investigated these infections in immunocompromised hosts, due to the high rate of progression to lower respiratory tract infections, and increased morbidity resulting in delay of chemotherapy [29]. Influenza accounted for up to 38% of these infections, similar to the rates in immunocompetent hosts during the influenza season [30, 31]. In a prospective study of viral respiratory infections in pediatric cancer patients at our center, we identified a respiratory virus in 87% of the participants, with RSV being the most prevalent, followed by parainfluenza and influenza B viruses [32].

Hospitalizations for influenza virus in the pediatric oncology patients are more frequent and range between two to seven days [33]. The mean length of hospital stay in our study was slightly higher, reaching 8 days. In pediatric oncology patients, intensive care admission rates increase to 10% of the influenza patients, and death occurs in up to 5% [34], similar to the study rates of 13% and 7% respectively. On the other hand, the burden of RSV in the pediatric population extends beyond infancy where it may cause bronchiolitis, bronchospasm, pneumonia and acute respiratory failure especially in children with co-morbidities with a mortality rate ranging between 1.6% in older children and 2.3% in neonates [35]. Although we encountered one mortality each in participants infected with RSV and influenza, it was difficult to attribute these mortalities to the viral infections because of the presence of other co-morbidities. Importantly studies have shown that the risk of RSV is higher in ALL patients with lower nadir absolute lymphocyte counts [31]; and profound lymphopenia is associated with a more severe course of illness [36]. Lymphopenia was present in 3 of the 7 participants with RSV.

In contrast to our study, which has identified only three respiratory viruses, previous studies have reported a wider array of different respiratory viruses. In a prospective study, Torres et. al reported a high prevalence of rhinovirus, followed by RSV, parainfluenza virus, and less commonly influenza virus, metapneumovirus and coronavirus [37]. Another retrospective study published recently from Italy also identified rhinovirus as the most common virus, followed by adenovirus, RSV and parainfluenza whereas influenza virus and metapneumovirus were less common [38]. The differences in distribution may be attributed to the different viral detection kits that were used: the PneumoVir multiplex PCR, the FilmArray Respiratory Panel, and PureLink Viral RNA/DNA mini Kit respectively in the studies and to differences in

inclusion criteria, seasonal variation, and population. In our institution on the other hand, the multiplex panel/multiplex ultra-panel were not introduced until early 2016, and even since have been minimally utilized in routine clinical care due to cost issues. Our cohort included only one coronavirus infection detected through these panels. This is likely applicable in other developing countries as well.

## GI infections

The most common GI infection studied in cancer patients is norovirus, which may cause severe gastrointestinal disease, more commonly in patients with HSCT [39]. Other viruses include sapovirus and astrovirus that are now being identified with the advent of new diagnostic techniques [39]. None of these viruses were identified in our study population, most likely due to under-diagnosis and under-referral for testing due to cost limitations. It may also be due to the fact that testing for rotavirus, bacterial and parasitic infections is more readily available. Multiplex PCR testing for GI infections is emerging as an important tool for earlier detection of infections. These panels usually test for a variety of bacterial, parasitic, and viral causes [40]. However, they were introduced only in the last year of our study and were underutilized due to their high cost.

Rotavirus is an important cause of viral gastroenteritis especially in children between the ages of six months and 2 years. Among immunocompromised children, even older participants with rotavirus infection may present with severe and prolonged gastroenteritis with high-grade fever, dehydration and acidosis [41]. The most common associated complications of rotavirus in the literature include necrotizing enterocolitis, intussusception, and seizures [42]. Although none of the participants in the study encountered these complications, one participant developed pancreatitis, an association that has been documented in only rare occasions [43].

## BK Virus infections

As for BK virus, the infection is typically acquired in early childhood, and by 10 years of age, 50% of children are seroconverted. After the primary infection, BK virus can be reactivated after latency in an immunocompromised host. Hemorrhagic cystitis occurs in HSCT patients and has a prevalence of about 10–25%. In other types of immunocompromised patients, hemorrhagic cystitis and nephropathy are rare [44], although the study cohort found to have BK virus had all presented with cystitis. Little information is found on the association of BKV with neutropenia and/or lymphopenia, due its rare occurrence in non-transplant patients, but it has been associated with myeloablative conditioning regimens used in HSCT, and concomitant CMV/adenovirus infections [45].

**Viral infections in relation to the underlying malignancy.** In contrast to the worldwide distribution of pediatric cancers, the most common pediatric cancer category in our cohort was solid tumors (52%), rather than leukemia (33%). This may have been due to referral bias since solid tumors are more likely to have complex treatment regimens that require referral to specialized center, such as CCCL. In this cohort, participants with leukemia had the majority of infections (58%). Due to the myelosuppressive-therapy regimens used in hematological malignancies, these patients are at high risk for various infections [46, 47]. Infections in solid tumors on the other hand, can also be secondary to procedures, barriers in the mucosa, and obstruction of natural passages [47].

**Viral infections in HSCT participants.** In cancer patients who undergo HSCT, the seropositivity of the donor and recipient affect the distribution of viral infections encountered. For example, CMV is more commonly seen post-transplant in patients who were seropositive for

CMV before the transplant leading to pre-emptive treatment [48]. In our study, the proportion of CMV infections in participants with HSCT was significantly higher than those without, but this may be due to the fact that CMV is routinely monitored in the former group and was incidentally detected. Other commonly encountered viruses in transplant patients such as VZV and HSV were also observed in our population with 70% detected within 6 weeks of the transplant, thus supporting anti-viral prophylaxis in the first few months post-transplantation [48].

**Limitations.** Very few studies provide a general overview of viral infections in the pediatric cancer population from developing countries. Despite filling some gaps, our study has several limitations. Being a retrospective study, cases included in our investigation were based on positive laboratory results for participants who were tested for certain viruses during their admission or based on clinical diagnosis (e.g. varicella). That being said, these viral studies were usually taken based on the treating team's degree of suspicion from the clinical presentation. A child presenting with respiratory symptoms during the winter season is more likely to be tested for RSV and influenza, while viruses such CMV and EBV are usually tested for after several days of failed antimicrobial therapy. As such, many viral infections might have been missed if the participants had presented without the appropriate clinical picture to raise suspicion for testing. Also, detected viruses may not always be responsible for the clinical presentation mentioned in our study, but this is usually left to the treating physician to decide whether the isolated virus is compatible with the respective clinical presentation. Furthermore, since the data was collected mostly before the introduction of multiplex PCR panels, many respiratory and gastrointestinal viral infections remained undiagnosed and thus the distribution of viruses may not provide a complete picture. The retrospective nature of the review dictates that the data may be incomplete and the documentation may be lacking, especially that pertaining to risk factors for infections and vaccination history. Additionally, the generalizability of the results is possibly limited by the unique distribution of viral agents in different areas of the world, though the overall distribution of viral categories may be comparable.

## Conclusion

The current study, despite its limitations, offers the first overview of viral infections in pediatric cancer patients from a developing country and will provide a useful reference for comparison for future studies performed after the adoption of more advanced diagnostic technologies. We observed that herpesviruses are the predominant group of viruses encountered, followed by respiratory and gastrointestinal viruses. Although viral infections probably contributed to the morbidity and mortality in our population, prospective studies in similar populations with systematic testing for different viruses under prespecified conditions and comparison of the infected versus uninfected participants would likely be more informative in delineating the true contribution to morbidity and mortality and in shaping the approach to infections in children and adolescents with cancer.

With the rising rates of antimicrobial resistance and the resulting efforts at antimicrobial stewardship, the identification of viral causes of infection can help limit the unnecessary and prolonged use of antibiotics in the pediatric cancer population especially for those presenting with fever and neutropenia. Testing for infection with most of the viruses reported in this study can be easily performed upon presentation. Thus, we recommend that physicians maintain a high degree of suspicion for viral infection and consider early testing in immunocompromised patients who present with compatible signs and symptoms in order to spare the use of antibiotics. With the introduction of more comprehensive virus PCR panels, we now have a powerful tool to detect viruses and the aim should be to make these panels more accessible and inexpensive for hospitals in developing countries. In addition, the difficulty encountered in

obtaining vaccination status from medical records for vaccine-preventable viral infections in the study such as VZV, rotavirus, and influenza highlights the importance of capturing this information in the record and engaging the primary care provider in the care of these patients. Finally, knowing that the threshhold for viral testing differs between institutions and even physicans in developing countries such as Lebanon, we believe it is important to set up clear protocols and algorithms that emphasize the importance of viral testing and treatment in immunocompromised patients admitted with a suspected infection.

## Acknowledgments

We would like to acknowledge Ms. Celina Boutros, BSN, and Ms. Mireille Lteif Khoury, BSN, for providing statistical advice and administrative support.

## Author Contributions

**Conceptualization:** Sarah Chamseddine, Ahmad Chmaisse, Imad Akel, Zeinab El Zein, Ghassan Dbaibo.

**Data curation:** Sarah Chamseddine, Ahmad Chmaisse, Imad Akel, Zeinab El Zein, Suzan Khalil, Sarah Abi Raad, Antoine Khati, Hiba Ghandour, Sarah Khafaja, Magda Haj.

**Formal analysis:** Sarah Chamseddine, Ahmad Chmaisse, Imad Akel, Sarah Khafaja.

**Investigation:** Sarah Chamseddine, Ahmad Chmaisse, Imad Akel, Zeinab El Zein, Suzan Khalil, Sarah Abi Raad, Antoine Khati, Hiba Ghandour, Sarah Khafaja, Miguel Abboud, Rami Mahfouz, George Araj, Hassan Zaraket, Ghassan Dbaibo.

**Methodology:** Sarah Chamseddine, Ahmad Chmaisse, Imad Akel, Rami Mahfouz, George Araj, Ghassan Dbaibo.

**Project administration:** Ahmad Chmaisse, Imad Akel, Ghassan Dbaibo.

**Resources:** Ghassan Dbaibo.

**Software:** Sarah Chamseddine, Ahmad Chmaisse, Sarah Khafaja.

**Supervision:** Sarah Chamseddine, Ahmad Chmaisse, Miguel Abboud, Rami Mahfouz, Rima Hanna-Wakim, Samar Muwakkit, Ghassan Dbaibo.

**Validation:** Sarah Chamseddine, Ahmad Chmaisse, Magda Haj, George Araj, Hassan Zaraket, Ghassan Dbaibo.

**Visualization:** Sarah Chamseddine, Ahmad Chmaisse, Ghassan Dbaibo.

**Writing – original draft:** Sarah Chamseddine, Ahmad Chmaisse, Suzan Khalil, Sarah Abi Raad, Antoine Khati, Hiba Ghandour, Sarah Khafaja, Miguel Abboud, Rima Hanna-Wakim, Samar Muwakkit, Ghassan Dbaibo.

**Writing – review & editing:** Ahmad Chmaisse, Miguel Abboud, Hassan Zaraket, Rima Hanna-Wakim, Samar Muwakkit, Ghassan Dbaibo.

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
