## [Decision Letter · Decision Letter 0]

15 Apr 2020

PONE-D-19-32741

Epidemiology and Clinical Characteristics of Viral Infections in Children and Adolescents with Cancer in a Developing Country

PLOS ONE

Dear Dr. Dbaibo,

Thank you for submitting your manuscript to PLOS ONE. After careful consideration, we feel that it has merit but does not fully meet PLOS ONE’s publication criteria as it currently stands. Therefore, we invite you to submit a revised version of the manuscript that addresses the points raised during the review process.

The Authors are expected to address all the criticisms by all Reviewers. In particular, please clarify the testing among the patients and potential bias due to lack of testing, revise the age group for adolescents and rewrite the manuscript for brevity and clarity (e.g. abstract results, results and discussion) (Reviewer #1 and #2), consider and discuss the potential heterogeneity in the patients (Reviewer #1) and clarify the discharge criteria (Reviewer #2). In additional to the above comments, please address,

Tables 3-6, please clarify the hypothesis being tested for the Fisher’s Exact test.Table 3. Why there was only 1 p-value?

We would appreciate receiving your revised manuscript by May 30 2020 11:59PM. To enhance the reproducibility of your results, we recommend that if applicable you deposit your laboratory protocols in protocols.io, where a protocol can be assigned its own identifier (DOI) such that it can be cited independently in the future. For instructions see: http://journals.plos.org/plosone/s/submission-guidelines#loc-laboratory-protocols

We look forward to receiving your revised manuscript.

Kind regards,

Eric HY Lau, Ph.D.

Academic Editor

PLOS ONE

Journal Requirements:

2. In ethics statement in the manuscript and in the online submission form, please provide additional information about the patient records used in your retrospective study. Specifically, please ensure that you have discussed whether all data were fully anonymized before you accessed them and/or whether the IRB or ethics committee waived the requirement for informed consent. If patients or parents of minors provided informed written consent to have data from their medical records used in research, please include this information.'

Additional Editor Comments (if provided):

The Authors are expected to address all the criticisms by all Reviewers. In particular, please clarify the testing among the patients and potential bias due to lack of testing, revise the age group for adolescents and rewrite the manuscript for brevity and clarity (e.g. abstract results, results and discussion) (Reviewer #1 and #2), consider and discuss the potential heterogeneity in the patients (Reviewer #1) and clarify the discharge criteria (Reviewer #2). In additional to the above comments, please address,

1. Table 3-6, please clarify the hypothesis being tested for the Fisher’s Exact test.

2. Table 3. Why there was only 1 p-value?

Reviewers' comments:

Reviewer's Responses to Questions

**Comments to the Author**

1. Is the manuscript technically sound, and do the data support the conclusions?

Reviewer #1: No

Reviewer #2: Yes

2. Has the statistical analysis been performed appropriately and rigorously? 

Reviewer #1: No

Reviewer #2: Yes

3. Have the authors made all data underlying the findings in their manuscript fully available?

Reviewer #1: No

Reviewer #2: Yes

4. Is the manuscript presented in an intelligible fashion and written in standard English?

Reviewer #1: No

Reviewer #2: Yes

5. Review Comments to the Author

Reviewer #1: Reviewer’s comment

This study conducted by Chamseddine et al. presented the “Epidemiology and clinical characteristics of viral infection in children and adolescents with cancer in Lebanon”. The authors included all kinds of detectable viral pathogens, compared their presentations and analyzed the association between different pathogens and underlying malignancies. Overall, the viral etiologies have different entry routes and pathogenesis in hosts with different immune status. It is a bit odd to compare all these together though at the same time those data might be valuable for clinicians in these regions when the guideline for treating neutropenic fever is to be established. However, analyzing data from a group of patients who did not receive the same tests could contribute to a great bias.

In result section, presenting cases with different viral infections read like case series/case report. The information should be more condensed and precise.

Article Title

Suggest to change “a developing country” to “Lebanon” to be more precise.

Materials and Methods

Page 6, Line 107-108

The defined age of adolescents by WHO is between 10-19 years of age. As the article title states “children and adolescents”, how did the authors decide to include subjects under 21 years of age?

Page 6, Line 112-114

“These viruses were detected using PCR, antibody detection, and/or antigen detection of samples taken from all body sources… ” To be more precise, viruses cannot be detected using antibody detection. The PCR methods for each pathogen should be stated clearly. Was it in-house PCR or done by commercial kits? Were different modalities of testing included in the same study? What kind of samples were used for what kind of tests should be stated more specifically as some viral detection from the throat swab might not be the true pathogens. Adopting results from different assays would introduce variation of sensitivity and specificity of the different tests.

Assuming testing BK virus is not routinely done, what is the indication for the clinicians to request BK virus test?

Results

Subject characteristics

Were all the patients hospitalized? If so, please include “hospitalized” in the article title.

It is indeed unusual to have more cases with solid tumor than leukemia. Even if there were fewer cases with leukemia, more episodes of viral infection occurred. This is as expected as the host immune status of those two disease entities is rather different. In my opinion, those 2 groups should be analyzed separately. It is surprising to see so low percentage of viral co-detection. The reason could be down to the detection method. If not all the patients receive the same tests, it is difficult to clarify how many of them have real co-infection or it was just because the test was not performed. Thus, using the same test modality for all the patients is suggested.

Viral infections in relation to the underlying malignancy

Page 10, Line 181-182

“No EBV infection were detected in subjects with solid tumors, while BK virus was only detected in those with solid tumors.” Did the authors perform tests for EBV and BK in all the patients? Did the authors use the same testing methods?”

Page 14, Line 258 onwards

VZV infection contains two clinical entities, including chickenpox and herpes zoster. This kind of viral infection is based on clinical diagnosis. It is important to clarify whether VZV infection is due to primary infection or reactivation.

Page 15, Line 270 onwards

It is surprising to see how low percentage of respiratory tract infection is. In Line 275, the authors mentioned the detection of coronavirus using multiplex PCR respiratory ultra-panel in only one case. Was the panel used for all the reported cases?

Page 29 Conclusion

The conclusion is too generalized. What is the specific contribution of this study? Do authors recommend routine screening of all viruses on hospitalized patients with fever? Or do authors suggest discontinuing antibiotics on all patients with positive evidence of viral infection? Please state clearly.

Table 1

Tested samples should be stated. BK virus was not included.

Table 5

Only if all the patients received all the tests to prove their positivity or negativity of the infection, there is no point of comparing the mixed results of the tested and untested.

Table 7

I noticed that the mortality of patients post HSCT was rather high. Do the authors think that the concurrent bacterial infection contributes more to the high mortality rate? If this is the case, what is the role of concurrent viral infection?

Reviewer #2: This is a very important paper that highlights the importance of viral infections in children with malignancy. Many times children who present with fever and other constitutional symptoms are thought to have to have bacterial infections and empirically treated with antibiotics, especially in low-income settings. The evidence provided in this manuscript that shows the spectrum of viral infections will be very helpful in raising the index of suspicion of viral aetiology, which if managed promptly and appropriately could improve patient outcomes.

Comments

In this retrospective study, the authors selected the records of the children who had been tested for viral infections as a possible cause of their symptoms or those with clear signs and symptoms of specific viral infections. It is not clear whether this is based on clinician's opinion or it is a routine practice in the study centre. Otherwise, it is possible that there might be some children with viral infections that were not part of the study because they had not been tested for.

Is there a specific discharge criteria in the centre, that was applied across all participants. This information is important in assessing the validity of the data on length of hospital stay. Otherwise, the differences could be due to the subjective discharge for clinicians and not necessarily the viral infections.

This study was about children and adolescents (as the title suggests). The WHO definition of an adolescent is any person between 10-19 years of age. It is not clear why the authors extended this to 21 years. Is this a country-specific definition?

Line 141-145: The authors indicate that for patients who were recurrently admitted with the same viral infections, only the first episode was considered as a separate case. It is not clear why this was the case, because, it is possible to have recurrent episodes of the same aetiological agent. Indeed, it would be important to further characterize this group f participants.

Results

The categorization of age is rather unusual. Generally, the Paediatric population is categorized as infants (under 12 months, although in some cases it is under 2 years, 2-5 years, 6-10 years, 10-19 years- the adolescents).

The titles for tables are usually at the top while for the figures, at the bottom. In this manuscript, all titles for tables are at the bottom. Is this the style of the journal?

Line 167-171 describes the population with febrile neutropenia, 6 of them with co-infection with bacteria. Did any of the 6 participants with co-infection have HSV, CMV or VZV? This is not clear.

I appreciate that this paper focused on epidemiology and clinical characteristics of viral infections. However, to improve the utility of the study results to a clinician in teasing out which children may have viral infections and therefore raise the index of suspicion and treat empirically especially in centres with poor diagnostic capacity, it would have been good to provide some information on the other children who did not have viral infections. What was the spectrum of the clinical and demographic characteristics? And outcomes? Would viral infections lead to prolonged hospital stay, or more complications?

The authors have done a good job in providing the detailed results for each group of viruses and the the systems they affect, including prevalence up to outcomes. However, one easily loses track of the message because the details are too many and some of them already provided in the tables.

Suggestion- the authors can identify the message for each of the results category and provide this in a summarized and logical fashion, to make it easier for the reader.

Discussion

The opening paragraph should provide a brief overview of the study objectives and key results to help the reader understand the discussion better. Secondly, it focuses on the proportions of the different tumors although the study is primarily about viral infections- this is the news and should come upfront.

Just like the narrative for the results, the discussion is quite lengthy and one easily gets lost in trying to understand the details of the discussion around each virus. The authors could consider identifying the message(s) they want to give the readers, and focus the discussion and conclusion there.

6. PLOS authors have the option to publish the peer review history of their article (what does this mean?). If published, this will include your full peer review and any attached files.

Reviewer #1: No

Reviewer #2: No

---

## [Author Response · Author response to Decision Letter 0]

31 May 2020

PONE-D-19-32741

Epidemiology and clinical characteristics of viral infections in hospitalized children and adolescents with cancer in Lebanon

PLOS ONE

Eric HY Lau, Ph.D. May 12, 2020

Academic Editor

PLOS ONE

Dear Dr. Lau,

We would like to thank you and the reviewers for taking the time to review our paper and provide us with constructive feedback. Thank you for finding that our manuscript has merit, and we hope that with the enclosed edits you will find it publishable in PLOS One. Below is a point-by-point response to the comments.

Editor’s Comments:

1) Materials and Methods: “In ethics statement in the manuscript and in the online submission form, please provide additional information about the patient records used in your retrospective study. Specifically, please ensure that you have discussed whether all data were fully anonymized before you accessed them and/or whether the IRB or ethics committee waived the requirement for informed consent. If patients or parents of minors provided informed written consent to have data from their medical records used in research, please include this information.”

Since our study was retrospective in nature, the Institutional Review Board (IRB) of our institution had waived the requirement for informed consent (PED.GD.07). Moreover, the data was anonymized at the analysis phase, as it was not possible to retrieve subjects’ data without accessing their charts using their Medical Records Number (MRN) at the collection phase. A section was added to clarify this matter in the “Study design” section (line 118).

2) Results: “Table 3-6, please clarify the hypothesis being tested for the Fisher’s Exact test.”

For reasons discussed in our reply to comment 3, please note that the p-value was removed from table 6 (previously table 3) as Reviewer #1 was not in favor of making the enclosed comparisons. We also chose to remove the p-value from table 4 (previously table 5), since subjects in this study were not tested for all viruses and the clinical presentation might not have been due to the detected virus solely, again as suggested by Reveiwer #1. The tested null hypothesis for the other tables was added under each table as follows:

- For table 3 (previously table 4) (line 213): There is no association between risk factors (chemotherapy, radiotherapy)/laboratory finding (neutropenia and Lymphopenia) and particular viruses, significant at p<0.05 Fisher’s exact test was used when expected count was less than 5.

- For table 5 (previously table 6) (line 230): There is no association between clinical outcomes (length of stay, ICU admission, intubation, complications, death) and particular viruses, significant at p<0.05. Fisher’s exact test was used when expected count was less than 5.

3) Results: “Table 3. Why there was only 1 p-value?”



The p-value in table 6 (previously table 3) was meant to show that a significant association exists between at least one virus and one of the malignancy groups. We decided to remove this p-value, as we agreed with Reviewer #1 that it would not be appropriate to assess the association between viruses among different malignancy groups given the heterogeneity among them.

Here, we would like to note that we had inadvertently omitted some of the authors’ names in our original submission. We updated the author names section to include all contributing authors.

Response to comments by Reviewer #1:

We thank reviewer #1 for the very precise observations, constructive comments, and helpful suggestions to improve our manuscript. Below is a point-by-point response to the comments.

1) Materials and Methods: “The authors included all kinds of detectable viral pathogens, compared their presentations and analyzed the association between different pathogens and underlying malignancies. Overall, the viral etiologies have different entry routes and pathogenesis in hosts with different immune status. It is a bit odd to compare all these together though at the same time those data might be valuable for clinicians in these regions when the guideline for treating neutropenic fever is to be established. However, analyzing data from a group of patients who did not receive the same tests could contribute to a great bias.”

The aim of our study is to give a general overview of viral infections encountered in children with cancer in a resource-limited country. This would help identify priorities for capacity building in countries with similar settings. Given the retrospective nature of our study, we were unable to ensure that all subjects had received the same tests, which we concede is a limitation. We only included subjects who were retrospectively found to have positive lab results for the included viruses, or who were discharged with a “discharge code” for a clincally-diagnosed virus. That being said, the specific viral laboratory studies are usually requested based on the treating team’s degree of suspicion from the symptomatology and clinical picture of the patient. For example, an infant presenting with respiratory symptoms is more likely to be tested for RSV and influenza, whereas a patient with persistent fever despite antibiotic and antifungal therapy gets tested for CMV and EBV. As such, testing was not uniform among included subjects and viral infections that were not tested for might have been missed. A statement was added to mention this in the “Limitations” section (line 515).

2) Results: “In result section, presenting cases with different viral infections read like case series/case report. The information should be more condensed and precise.”

Thank you for highlighting this point. In the accompanying draft, we have attempted to summarize and present the information in a more condensed and precise manner.

3) Article Title: “Suggest to change “a developing country” to “Lebanon” to be more precise.”.

As advised, we have changed the article’s title in order to more accurately describe our paper. The title is now “Epidemiology and Clinical Characteristics of Viral Infections in Hospitalized Children and Adolescents with Cancer in Lebanon”.

4) Materials and Methods: “Page 6, Line 107-108. The defined age of adolescents by WHO is between 10-19 years of age. As the article title states “children and adolescents”, how did the authors decide to include subjects under 21 years of age?”

The Children’s Cancer Center of Lebanon (CCCL) admits and treats all patients between the ages of 0 and 21. Knowing this and given the overlap between younger adolescents’ (14-16 years old) cancers/treatments and older adolescents’/young adults’ (20-21 years old), the adolescent age limit was considered up until 21 years of age (similar to American Academy of Pediatrics upper age limit of adolescents), instead of WHO’s upper age limit.

5) Materials and Methods: ““These viruses were detected using PCR, antibody detection, and/or antigen detection of samples taken from all body sources… ” To be more precise, viruses cannot be detected using antibody detection. The PCR methods for each pathogen should be stated clearly.”

a) Materials and Methods: “Was it in-house PCR or done by commercial kits?” 

PCR commercial kits were used for all viruses. A column was added in Table 1 to indicate the commercial kits used.

b) Materials and Methods: “Were different modalities of testing included in the same study?”

Common viral detection methods utilized at our center are mentioned in table 1 and were consistent for the duration of the study. Overall, the same modalities for sampling and diagnosis were used for individual viruses as indicated in the modified Table 1, which now includes the number of cases diagnosed by each modality. 

c) Materials and Methods: “What kind of samples were used for what kind of tests should be stated more specifically as some viral detection from the throat swab might not be the true pathogens.”

Table 1 was expanded to include this information. We agree with the reviewer that detecting some viruses in the throat may not always prove their responsibility for the clinical presentation. This is usually left to the treating physician to decide whether the isolated virus is compatible with the respective clinical presentation. We include this qualifying statement in the limitations (line 523).

d) Materials and Methods: “Adopting results from different assays would introduce variation of sensitivity and specificity of the different tests.”

We fully agree with the reviewer on this point. Unfortunately, we were unable to avoid the variation in sensitivity and specificity of the different tests as this was up to the treating physician to decide which test to use. The reason we had relied on these different methods was to avoid missing any viruses detected, and only positive results were reported and discussed. Fortunately, for the majority of the viruses the same testing modality was used throughout as summarized in Table 1 with only two exceptions (adenovirus 3 cases, and CMV one case diagnosed by positive IgM and the remaining 22 by PCR).

6) Materials and Methods: “Assuming testing BK virus is not routinely done, what is the indication for the clinicians to request BK virus test?”

Patients receiving chemotherapy who presented with signs and symptoms of “hemorrhagic cystitis” (hematuria, dysuria, vague abdominal pain…) were tested of BK virus in their urine. This was confirmed in the charts of subjects who presented first with such symptoms and were later found to be positive for BK virus as indicated (line 301).

7) Results: “Were all the patients hospitalized? If so, please include “hospitalized” in the article title.”

All patients were hospitalized. As advised, we have changed the article’s title to more accurately describe our paper. The title is now “Epidemiology and Clinical Characteristics of Viral Infections in Hospitalized Children and Adolescents with Cancer in Lebanon”.

8) Results: “It is indeed unusual to have more cases with solid tumor than leukemia. Even if there were fewer cases with leukemia, more episodes of viral infection occurred. This is as expected as the host immune status of those two disease entities is rather different. In my opinion, those 2 groups should be analyzed separately. It is surprising to see so low percentage of viral co-detection. The reason could be down to the detection method. 

Unfortunately, it was difficult to analyze different malignancy groups separately, as it would lead to low numbers in each group and inability to derive meaningful results. For this reason, we focused instead on providing an overview of common viruses in all pediatric cancer subjects.

9) If not all the patients receive the same tests, it is difficult to clarify how many of them have real co-infection or it was just because the test was not performed. Thus, using the same test modality for all the patients is suggested.”

As indicated in the answer to question 1, only subjects who were retrospectivelly found to have a positive viral infection in their laboratory findings were included. For this reason, viral co-inefctions were less likely to be detected if the patient was not tested for other viruses.

10) Results: “Viral infections in relation to the underlying malignancy. Page 10, Line 181-182. No EBV infection were detected in subjects with solid tumors, while BK virus was only detected in those with solid tumors. Did the authors perform tests for EBV and BK in all the patients? Did the authors use the same testing methods?”

The decision to test for EBV or BK viruses was made by the treating team. As mentioned above, testing for EBV is usually done in the context of prolonged fever not responding to antimicrobial treatment whereas testing for BK virus is done when there is clinical and laboratory evidence of hemorrhagic cystitis. Since our retrospective study only captured the positive cases, there is no way for us to determine the number of patients who were tested and were negative for either virus. Certainly, not all patients were tested as this was not a systematic prospective study but rather a retrospective observation that reflects the practices of treating physicians. 

11) Results: “Page 14, Line 258 onwards. VZV infection contains two clinical entities, including chickenpox and herpes zoster. This kind of viral infection is based on clinical diagnosis. It is important to clarify whether VZV infection is due to primary infection or reactivation.”

26 (60%) subjects presented with a primary chickenpox infection, while 17 (40%) subjects presented with a reactivated VZV infection. This was clarified in the section discussing VZV infections (line 258) 

12) Results: “In Line 275, the authors mentioned the detection of coronavirus using multiplex PCR respiratory ultra-panel in only one case. Was the panel used for all the reported cases?”

Unfortunately, the respiratory panel was introduced in mid 2017 to AUBMC (towards the tail end of our study) at a high cost. Due to cost-containment strategies at the Children’s Cancer Center of Lebanon, it was not widely used, which might explain the low detection rates of respiratory viruses other than influenza and RSV (this was mentioned in the limitations section, line 542). With regards to the coronavirus case, it was the only virus detected through this respiratory panel during our study period.

13) Conclusion: “The conclusion is too generalized. What is the specific contribution of this study? Do authors recommend routine screening of all viruses on hospitalized patients with fever? Or do authors suggest discontinuing antibiotics on all patients with positive evidence of viral infection? Please state clearly.”

Thank you for pointing this out to us. We expanded our conclusion and gave more specific recommendations for physicians and researchers (line 537).

14) Results: “Table 1: Tested samples should be stated. BK virus was not included.”

Tested samples were stated in Table 1 and BK Virus was added to this table.

15) Results: “Table 5: Only if all the patients received all the tests to prove their positivity or negativity of the infection, there is no point of comparing the mixed results of the tested and untested.”

As per your comment, we removed the p-value from table 4 (line 216), previously table 5, considering the symptomatology could have been caused by another untested virus. 

16) Table 7: I noticed that the mortality of patients post HSCT was rather high. Do the authors think that the concurrent bacterial infection contributes more to the high mortality rate? If this is the case, what is the role of concurrent viral infection?

Given the low number of HSCT subjects and mortalities (only 3 deaths), it was difficult to draw conclusions regarding factors leading to this relative higher mortality percentage. It is difficult to attribute the mortality in HSCT patients to a single cause. These patients develop multiple infectious and non-infectious morbidities, the sum of which contributes to the mortality when it occurs. We cannot ascertain the relative contribution of viral versus bacterial infection and we would hesitate to make a conclusion based on our data. We have also clarified the table to indicate that the number is 3 and the percentage is 11.

 

Response to the comments by Reviewer #2:

We thank this reviewer for indicating that our paper is “very important, …especially in low-income settings” and that it “shows the spectrum of viral infections will be very helpful in raising the index of suspicion of viral aetiology, which if managed promptly and appropriately could improve patient outcomes.” Below is a point-by-point response to the comments.

1) Materials and Methods: “In this retrospective study, the authors selected the records of the children who had been tested for viral infections as a possible cause of their symptoms or those with clear signs and symptoms of specific viral infections. It is not clear whether this is based on clinician's opinion or it is a routine practice in the study center. Otherwise, it is possible that there might be some children with viral infections that were not part of the study because they had not been tested for.”

Thank you for highlighting this very important point. This represents one of the limitations of our retrospective study. The testing for specific viral infections was decided on by the treating team. This is usually driven by the clinical presentation and its evolution during hospitalization. For example, a child with fever and neutropenia not responding to antimicrobial treatment may be tested for EBV and CMV. Another child, with respiratory symptoms during the winter season may be tested for RSV and influenza. However, there is no routine practice of testing for all the viruses of interest due to the limited resources at our center. The cases captured in our study were those whose testing results turned out positive for specific viruses or were clinically diagnosed with specific viruses (e.g. varicella). We were unable to determine what fraction of other patients were tested and had negative results. We agree that it is very likely that patients with viral infections were missed due to the lack of testing. The answer to this question would require a large prospective study with systematic testing under prespecified conditions. A section was added to mention this in the “Limitations” section (line 515).

2) Materials and Methods: “Is there a specific discharge criteria in the centre, that was applied across all participants. This information is important in assessing the validity of the data on length of hospital stay. Otherwise, the differences could be due to the subjective discharge for clinicians and not necessarily the viral infections.”

At the Children’s Cancer Center of Lebanon (CCCL), patients are generally discharged when they are clinically better and are able to continue their treatment at home, if necessary. This is mostly driven by cost-saving on this charity-run center and all physicians abide by this goal leaving little room for variation between physicians. 

3) Materials and Methods: “This study was about children and adolescents (as the title suggests). The WHO definition of an adolescent is any person between 10-19 years of age. It is not clear why the authors extended this to 21 years. Is this a country-specific definition?”

The Children’s Cancer Center of Lebanon (CCCL) admits and treats all patients between the ages of 0 and 21. Knowing this and given the overlap between younger adolescents’ (14-16 years old) cancers/treatments and older adolescents’/young adults’ (20-21 years old), the adolescent age limit was considered up until 21 years of age (similar to the American Academy of Pediatrics upper age limit of adolescents), instead of WHO’s upper age limit.

4) Materials and Methods: “Line 141-145: The authors indicate that for patients who were recurrently admitted with the same viral infections, only the first episode was considered as a separate case. It is not clear why this was the case, because, it is possible to have recurrent episodes of the same aetiological agent. Indeed, it would be important to further characterize this group of participants.”

Viruses such as CMV, EBV, and Adenovirus are known to persist for longer periods of time in immunocompromised subjects, and are usually closely followed for resolution of the virus. For this reason, and to avoid reporting the same persisting viral infection as multiple infections, a decision was taken to only report this as one viral infection. For other scenarios where patients have re-infection with the same virus after initial resolution, we specificed in the “Hospital admission and course of infection” section that “Subjects who had complete resolution of the infection, whether clinically and/or laboratory-proven, with recurrence of the same viral infection within one year were included as separate cases.”Line 158

5) Results: “The categorization of age is rather unusual. Generally, the Paediatric population is categorized as infants (under 12 months, although in some cases it is under 2 years, 2-5 years, 6-10 years, 10-19 years- the adolescents).”

We have edited the categorization of age in table 2 to better describe our patient population’s age distribution as suggested by the reviewer.

6) Results: “The titles for tables are usually at the top while for the figures, at the bottom. In this manuscript, all titles for tables are at the bottom. Is this the style of the journal?”

Thank you for pointing this out to us. We have fixed the tables to comply with the journal’s style.

7) Results: “Line 167-171 describes the population with febrile neutropenia, 6 of them with co-infection with bacteria. Did any of the 6 participants with co-infection have HSV, CMV or VZV?”

Among the subjects who presented with febrile neutropenia, the following were the cases of viral infection that occurred concurrently with a bacterial infection:

- 1 EBV case

- 2 HSV cases

- 1 VZV case

- 1 influenza case

- 1 CMV case

This was also clarified in the “Subject Characteristics and Detected Viruses” section (line 204).

8) Results: “I appreciate that this paper focused on epidemiology and clinical characteristics of viral infections. However, to improve the utility of the study results to a clinician in teasing out which children may have viral infections and therefore raise the index of suspicion and treat empirically especially in centres with poor diagnostic capacity, it would have been good to provide some information on the other children who did not have viral infections. What was the spectrum of the clinical and demographic characteristics? And outcomes? Would viral infections lead to prolonged hospital stay, or more complications?”

We agree with the reviewer that it would have been helpful to compare outcomes between children with viral infections with those without. Unfortunately, given the retrospective nature of our study, this is impossible as not all pediatric cancer patients admitted to the hospital during the study’s time period were prospectively screened for viruses. As such, we believed it would be inappropriate to compare the current subjects, who were either found positive for the virus by testing or had a viral-related discharge diagnosis, with subjects who might or might not have been tested and could have been infected with any of the viruses during their admissions.

9) Results: “The authors have done a good job in providing the detailed results for each group of viruses and the systems they affect, including prevalence up to outcomes. However, one easily loses track of the message because the details are too many and some of them already provided in the tables. Suggestion- the authors can identify the message for each of the results category and provide this in a summarized and logical fashion, to make it easier for the reader.”

Thank you for highlighting this point. In the accompanying draft, we have attempted to summarize and present the information in a more condensed and precise manner.

10) Discussion: “The opening paragraph should provide a brief overview of the study objectives and key results to help the reader understand the discussion better. Secondly, it focuses on the proportions of the different tumors although the study is primarily about viral infections- this is the news and should come upfront.”

An opening paragraph was added in the “Discussion” (line 380). With regards to the initial presentation of the proportion of tumors versus the viral infections, we re-organized both the “Results” and the “Discussion” sections to present the viral infections first, before their distribution among the different tumors.

11) Discussion: “Just like the narrative for the results, the discussion is quite lengthy and one easily gets lost in trying to understand the details of the discussion around each virus. The authors could consider identifying the message(s) they want to give the readers, and focus the discussion and conclusion there.”

In the accompanying draft, we have attempted to summarize and present the information in a more condensed and precise manner.

---

## [Decision Letter · Decision Letter 1]

3 Aug 2020

PONE-D-19-32741R1

Epidemiology and Clinical Characteristics of Viral Infections in Hospitalized Children and Adolescents with Cancer in Lebanon

PLOS ONE

Dear Dr. Dbaibo,

Thank you for submitting your manuscript to PLOS ONE. After careful consideration, we feel that it has merit but does not fully meet PLOS ONE’s publication criteria as it currently stands. Therefore, we invite you to submit a revised version of the manuscript that addresses the points raised during the review process.

The Authors are expected to address all the criticisms by all Reviewers. In particular, please revise the abstract conclusion on the impact of viral infections on mortality and morbidity and discussion the impact of vaccination (or lack of information) (Reviewer #2). In additional to the above comments, please address,

Tables 4, 6, please remove Fisher’s Exact test in the footnote. You may also indicate why the test was not carried out.

We look forward to receiving your revised manuscript.

Kind regards,

Eric HY Lau, Ph.D.

Academic Editor

PLOS ONE

Journal Requirements:

Additional Editor Comments (if provided):

The Authors are expected to address all the criticisms by all Reviewers. In particular, please revise the abstract conclusion on the impact of viral infections on mortality and morbidity and discussion the impact of vaccination (or lack of information) (Reviewer #2). In additional to the above comments, please address,

1. Tables 4, 6, please remove Fisher’s Exact test in the footnote. You may also indicate why the test was not carried out.

Reviewers' comments:

Reviewer's Responses to Questions

**Comments to the Author**

1. If the authors have adequately addressed your comments raised in a previous round of review and you feel that this manuscript is now acceptable for publication, you may indicate that here to bypass the “Comments to the Author” section, enter your conflict of interest statement in the “Confidential to Editor” section, and submit your "Accept" recommendation.

Reviewer #2: (No Response)

2. Is the manuscript technically sound, and do the data support the conclusions?

Reviewer #2: Yes

3. Has the statistical analysis been performed appropriately and rigorously? 

Reviewer #2: Yes

4. Have the authors made all data underlying the findings in their manuscript fully available?

Reviewer #2: Yes

5. Is the manuscript presented in an intelligible fashion and written in standard English?

Reviewer #2: Yes

6. Review Comments to the Author

Reviewer #2: The authors have made significant improvements on the manuscript.

A few comments below;

1. In the conclusion, the authors state that viral infections contributed significantly to morbidity and mortality. It is difficult to conclude that this was the case without data on the same variable in a comparison group (those with no viral infections). It is indeed possible that the morbidity and mortality was comparable in both groups.

In addition, there is no definite evidence from the data that the cause of death in these patients was the viral infections. Cancer patients commonly have multimorbidity, and potential multiple causes of death. It might be better to make these statements in a probabilistic manner.

2. The terms patients and subjects are used interchangeably. Ethically, it is better to refer to them as participants. Once an individual is on a study, even retrospectively, they are referred to as participants, and this term needs to be used wherever appropriate.

3. Line 619-626 has scientific facts, but it is not clear how they relate to the current study. Were there any participants who had the complications mentioned? If so, state that and discuss that result.

4. The results showed that it was not possible to get a record of the participants’ vaccination status. I propose that make note of this and discuss the wide implications in terms of general child health but also in the context of cancer patients, and probably a recommendation on improvements on data capture on vaccination.

5. In the conclusion, the authors make a note of the importance of rational antibiotic use which can be guided by a high index of suspicion of viral infections by the clinicians. However, they backtrack on this important issue by suggesting that this can happen after failure of broad-spectrum and this defeats the argument on rational antibiotic use.

6. Generally, the conclusion section seem more focused on recommendations (which is fine) but I miss specific concluding statements related to the study objective (s) and results.

7. PLOS authors have the option to publish the peer review history of their article (what does this mean?). If published, this will include your full peer review and any attached files.

Reviewer #2: **Yes: **Rebecca Nantanda

---

## [Author Response · Author response to Decision Letter 1]

16 Aug 2020

PONE-D-19-32741R1

Epidemiology and clinical characteristics of viral infections in hospitalized children and adolescents with cancer in Lebanon

PLOS ONE

Eric HY Lau, Ph.D. August 16, 2020

Academic Editor

PLOS ONE

Dear Dr. Lau,

We would like to thank you and reviewer #2 for taking the time to review our paper and provide us with additional constructive feedback. Thank you for finding that our manuscript has merit, and we hope that with the enclosed edits you will find it publishable in PLOS One. Below is a point-by-point response to the comments.

Editor’s Comment:

“Tables 4, 6, please remove Fisher’s Exact test in the footnote. You may also indicate why the test was not carried out.”

We have removed the Fisher’s Exact test in the footnote as suggested.

We thank Reviewer #2 for her thoughtful and helpful comments. Please find below the comments and their respective responses: 

1. In the conclusion, the authors state that viral infections contributed significantly to morbidity and mortality. It is difficult to conclude that this was the case without data on the same variable in a comparison group (those with no viral infections). It is indeed possible that the morbidity and mortality was comparable in both groups.

In addition, there is no definite evidence from the data that the cause of death in these patients was the viral infections. Cancer patients commonly have multimorbidity, and potential multiple causes of death. It might be better to make these statements in a probabilistic manner.

Response: We agree with the reviewer and have changed the wording of the opening sentence of the abstract conclusion to indicate that viral infections “may” have contributed to morbidity and mortality. 

2. The terms patients and subjects are used interchangeably. Ethically, it is better to refer to them as participants. Once an individual is on a study, even retrospectively, they are referred to as participants, and this term needs to be used wherever appropriate.

Response: We thank the reviewer for this suggestion. We replaced all "subjects" and most “patients” with "participants" with the exception of text referring to patients prior to them becoming participants, which were referred to as "patients" (i.e. while describing methods and selection process)

3. Line 619-626 has scientific facts, but it is not clear how they relate to the current study. Were there any participants who had the complications mentioned? If so, state that and discuss that result.

Response: We thank the reviewer for this comment. The two references relating to secondary bacterial complications of influenza are not immediately relevant to our findings, so we decided to eliminate them. We have modified the language of these lines to read as follows: “On the other hand, the burden of RSV in the pediatric population extends beyond infancy where it may cause bronchiolitis, bronchospasm, pneumonia and acute respiratory failure especially in children with co-morbidities with a mortality rate ranging between 1.6% in older children and 2.3% in neonates (35). Although we encountered one mortality each from RSV and influenza in our participants, it was difficult to attribute these mortalities to the viral infections because of the presence of other co-morbidities”.

4. The results showed that it was not possible to get a record of the participants’ vaccination status. I propose that make note of this and discuss the wide implications in terms of general child health but also in the context of cancer patients, and probably a recommendation on improvements on data capture on vaccination.

Response: We thank the reviewer for this suggestion. We have emphasized this in the “Limitations” and included the following statement in the “Conclusion”: “In addition, the difficulty encountered in obtaining vaccination status from medical records for vaccine-preventable viral infections in the study such as VZV, rotavirus, and influenza highlights the importance of capturing this information in the record and engaging the primary care provider in the care of these patients”. 

5. In the conclusion, the authors make a note of the importance of rational antibiotic use which can be guided by a high index of suspicion of viral infections by the clinicians. However, they backtrack on this important issue by suggesting that this can happen after failure of broad-spectrum and this defeats the argument on rational antibiotic use.

Response: We thank the reviewer for pointing this out and we agree that this defeats the argument. We have changed the wording of the relevant sentences in the “Conclusion” to the following: “With the rising rates of antimicrobial resistance and the resulting efforts at antimicrobial stewardship, the identification of viral causes of infection can help limit the unnecessary and prolonged use of antibiotics in the pediatric cancer population especially for those presenting with fever and neutropenia. Testing for infection with most of the viruses reported in this study can be easily performed upon presentation. Thus, we recommend that physicians maintain a high degree of suspicion for viral infection and consider early testing in immunocompromised patients who present with compatible signs and symptoms in order to spare the use of antibiotics”.

6. Generally, the conclusion section seems more focused on recommendations (which is fine) but I miss specific concluding statements related to the study objective (s) and results.

Response: We thank the reviewer for pointing this omission. We have restructured the “Conclusion” to highlight the findings from study in the beginning and to provide the recommendations afterwards.

For the authors,

Ghassan Dbaibo, M.D., FAAP

Professor and Vice-Chair for Research and Faculty Development

Department of Pediatrics and Adolescent Medicine

Head, Division of Pediatric Infectious Diseases

Director, Center for Infectious Diseases Research

Co-Director, Primary Immunodeficiency Diseases Program

Professor, Department of Biochemistry and Molecular Genetics

American University of Beirut

PO Box 11-0236

Riad El-Solh

Beirut, Lebanon

Tel: +961-1374374 

ext: 5540 Hospital Office

ext: 5752 Clinical Research Office

ext: 4819 Biochemistry

Fax: +961-1-370781

gdbaibo@aub.edu.lb

---

## [Editor Report · Decision Letter 2]

3 Sep 2020

Epidemiology and Clinical Characteristics of Viral Infections in Hospitalized Children and Adolescents with Cancer in Lebanon

PONE-D-19-32741R2

Dear Dr. Dbaibo,

We’re pleased to inform you that your manuscript has been judged scientifically suitable for publication and will be formally accepted for publication once it meets all outstanding technical requirements.

Kind regards,

Eric HY Lau, Ph.D.

Academic Editor

PLOS ONE
---

## [Editor Report · Acceptance letter]

11 Sep 2020

PONE-D-19-32741R2 

Epidemiology and clinical characteristics of viral infections in hospitalized children and adolescents with cancer in Lebanon 

Dear Dr. Dbaibo:

I'm pleased to inform you that your manuscript has been deemed suitable for publication in PLOS ONE. Congratulations! Your manuscript is now with our production department. 

Kind regards, 

on behalf of

Dr. Eric HY Lau 

Academic Editor

PLOS ONE